# Recent Developments and Future Perspectives of Vaccines and Therapeutic Agents against SARS-CoV2 Using the BCOV_S1_CTD of the S Protein

**DOI:** 10.3390/v15061234

**Published:** 2023-05-24

**Authors:** Amit Gupta, Ashish P. Singh, Vinay K. Singh, Rajeshwar P. Sinha

**Affiliations:** 1Laboratory of Photobiology and Molecular Microbiology, Centre of Advanced Study in Botany, Institute of Science, Banaras Hindu University, Varanasi 221005, India; amitgupta.bhu15@gmail.com (A.G.); singhashishpratap24@gmail.com (A.P.S.); 2Centre for Bioinformatics, School of Biotechnology, Institute of Science, Banaras Hindu University, Varanasi 221005, India; vinaysingh@bhu.ac.in; 3University Center for Research & Development (UCRD), Chandigarh University, Chandigarh 140413, India

**Keywords:** SARS-CoV-2, antiviral agents, Omicron variant, mutant, vaccines, clinical studies

## Abstract

Since the onset of the coronavirus disease 2019 (COVID-19) pandemic, the virus kept developing and mutating into different variants over time, which also gained increased transmissibility and spread in populations at a higher pace, culminating in successive waves of COVID-19 cases. The scientific community has developed vaccines and antiviral agents against severe acute respiratory syndrome coronavirus 2 (SARS-CoV-2) disease. Realizing that growing SARS-CoV-2 variations significantly impact the efficacy of antiviral therapies and vaccines, we summarize the appearance and attributes of SARS-CoV-2 variants for future perspectives in drug design, providing up-to-date insights for developing therapeutic agents targeting the variants. The Omicron variant is among the most mutated form; its strong transmissibility and immune resistance capacity have prompted international worry. Most mutation sites currently being studied are in the BCOV_S1_CTD of the S protein. Despite this, several hurdles remain, such as developing vaccination and pharmacological treatment efficacies for emerging mutants of SARS-CoV-2 strains. In this review, we present an updated viewpoint on the current issues faced by the emergence of various SARS-CoV-2 variants. Furthermore, we discuss the clinical studies conducted to assist the development and dissemination of vaccines, small molecule therapeutics, and therapeutic antibodies having broad-spectrum action against SARS-CoV-2 strains.

## 1. Introduction

SARS-CoV-2 is highly infectious, and it was the causative agent of the outbreak of the COVID-19 disease in 2019. The WHO declared it a global pandemic [1]. More than 676 million cases have been reported worldwide, with over 6.88 million deaths since late 2019 [2]. In the current situation, COVID-19 continues to endanger human welfare and impose unquestionable costs on healthcare. A reasonable strategy for lowering the mortality brought on by viral lung penetration is to neutralize SARS-CoV-2 before it enters human cells [3]. It has been accepted that vaccination-induced immunity is essential for controlling the spread of COVID-19. Because many recently discovered variations are immune-evading [4], novel treatments that prevent viral cell entrance are required as supplementary options.

The cryo-electron microscopy structure of the SARS-CoV-2 spike trimer has just been published in two separate investigations [5,6]. However, a closer look at one of the available spike structures indicated that the RBD was only partially modeled, specifically for the receptor-binding motif (RBM) that directly interacts with ACE2. The interaction with the N-terminal peptidase domain of ACE2 is specifically facilitated by the Betacoronavirus spike (S) glycoprotein S1 subunit C-terminal (BCOV_S1_CTD) [7]. Therefore, the sequence and structural analysis of BCOV_S1_CTD in the major variations was carried out for more information. In the site analysis of the sequence alignment, 89.11% conserved sites and 10.88% variable sites were identified. Based on this investigation, nearly 10.88% of sites played an important role in amino acid substitution. The spike glycoprotein (S protein) structure is likely to change because of these mutations [8]. The receptor-binding domain (RBD) is a primary target of efficient neutralizing antibodies, according to earlier investigations on SARS-CoV. However, it is necessary to consider whether these alterations impact the antigenicity of S proteins and their capacity to bind neutralizing antibodies. If the S protein’s B-cell epitopes are altered and could no longer bind to neutralizing antibodies, then the developed vaccines (based on prototype S protein) lose their effectiveness [9]. Here, we share the immuno-bioinformatic resources from the IEDB and related resources that were used to predict the B-cell epitopes in the BCOV_S1_CTD of the S protein from the major variants and compare the changes in the likely epitope sites from dominant and extremely rare mutations of the S protein. The Omicron variant and its subvariant have been found to have multiple mutations simultaneously [10]. We observed that different S proteins’ BCOV_S1_CTD mutations might have varying effects on the proteins’ putative functional epitopes. Access to affordable and reliable vaccinations is now a significant problem with vaccine usage. The theory underlying the design of the three primary vaccine types (protein subunit, adenoviral vector, and mRNA) and the effectiveness of vaccinations against various SARS-CoV-2 variants are discussed in subsequent sections.

Furthermore, the most optimistic possibility of stopping the pandemic is a successful rollout of COVID-19 immunization. SARS-CoV-2 continues to evolve, posing a greater risk to public health globally due to its quicker transmission and increased infectivity efficiency. The WHO categorizes them into variations of interest (VOIs) and variants under monitoring (VUMs) to analyze the effects of various variants and promote preventative or medicinal countermeasures more accurately.

## 2. The SARS-CoV-2 Structure

SARS-CoV-2, a member of the b-CoV class of human coronaviruses, is an enclosed virus with a diameter ranging from 80 to 220 nm, with positive single-stranded RNA within its shell [11]. Four structural proteins, delicate lipid envelopes, and genomic RNA comprise the complete SARS-CoV-2 particle [12]. The membrane protein (M), nucleocapsid protein (N), spike protein (S), and envelop protein (E) are the four structural proteins. In the replication process of viruses, the M protein is essential [12,13]. Its presence makes viruses and host components possible to assemble on the cell membrane to create progeny viral particles [14]. In viral transcription and assembly, the complex produced by the N protein and genomic RNA is crucial. The N-terminal, C-terminal, and disordered central regions of the N protein are known as the NTD, CTD, and RNA binding domains [15]. Table 1 displays the coronavirus spike glycoprotein domains’ profile from SPIKE_SARS2. The SARS-CoV-2 E protein is a very small, fully functional membrane protein that participates in various viral life cycle processes, including pathogenicity and assembly. For SARS-CoV-2 to enter cells, the S protein, which is present as trimers on the viral membrane surface, is crucial [16]. It comprises the S2 subunit, the spike protein’s most conserved structural component [16]. On the surface of the viral membrane, the E protein and the M protein are sequentially organized. The N protein interacts with the viral RNA to create the virus particle’s core, while the S protein builds the virus particle.

### 2.1. SARS-CoV-2 S Protein Attached to ACE2

The homotrimeric spike glycoprotein on the envelope of coronaviruses, which consists of an S1 subunit and an S2 subunit in each spike monomer, binds to the cellular receptors. Such binding sets off a series of processes that result in viral and cell membranes fusing for cell penetration. The SARS-CoV spike protein’s interaction with the cell receptor ACE2 has previously been studied using cryo-electron microscopy. These studies have revealed that receptor binding causes S1 and ACE2 to dissociate, which causes S2 to transition from the metastable pre-fusion state to an even more stable post-fusion state necessary for membrane fusion [17]. To infect target cells, SARS-CoV must first bind to the ACE2 receptor, which is a crucial first step. X-ray crystallography was used to establish the structure of the BCOV_S1_CTD-ACE2 complex in order to better comprehend the interaction between BCOV_S1_CTD and ACE2. Particularly, we observed an interaction between the N-terminal peptidase domain of ACE2 (residues Ser19–Asp615) and the C-terminal portion of the Betacoronavirus spike (S) glycoprotein S1 subunit (BCOV_S1_CTD) (residues 334–527) (Figure 1A–C). The peptide substrate-binding region is formed by the interaction of two lobes in the N-terminal peptidase domain of ACE2. The bottom side of the minor lobe of ACE2 is in contact with the expanded RBM in BCOV_S1_CTD, which has a concave outer surface (residues S438–Q506) that can be incorporated into the N-terminal helix of ACE2 (Figure 1C).

The VADAR analysis prediction of BCOV_S1_CTD revealed 6% helix, 77% beta, 53% coil, and 56% turns, with mean H-bond energy of −1.6 (SD = 1.1) against the expected value of −2.0 (SD = 0.8) in the protein [18]. As measured by several servers, these parameters indicated that the model was efficient, and structural stability was maintained, thereby validating the structure. The PDBsum tool was used to analyze the secondary structural elements more intensively. Here, the amino acid sequence of BCOV_S1_CTD was the secondary structure displaying seven helices (H1–H7), while β-sheet motifs were composed of 13 β-strands, 26 betas, and 3 gamma turns (Figure 1B).

### 2.2. Multiple Sequence and Structure Alignment of BCOV_S1_CTD

The S-protein sequence retrieval and identification of the BCOV_S1_CTD conserved domain region of all major variants were carried out using the UniProtKB database and PROSITE server. FASTA protein sequences were used for multiple sequence alignment (MSA). CLUSTAL OMEGA-based sequence alignment was used to identify the conserved region and mutation among the sequences of major variants [19]. The SALIGN web server was used to determine the best alignment procedure based on the inputs while allowing the user to override default parameter values [20]. Multiple sequence alignments were guided by a dendrogram computed based on a matrix of all pairwise alignment scores. Furthermore, the MEGA 7 tool was used to calculate conserved, variable, passim-informative, and singleton sites [19].

In the site analysis of the sequence alignment, 89.11% conserved sites, 10.88% variable sites, 9.3% passim-informative sites, and 1.5% singleton sites were identified. Based on this investigation, nearly 10.88% of sites played an important role in amino acid substitution. The conserved and consensus sequence is shown in Figure 2A. Figure 2A shows a black region in the conserved region showing substitution within major variants. The phylogenetic tree was constructed based on alignment data, and it was found that two major clusters were formed. The first cluster contained Omicron variants and differed from the second cluster (Figure 2B,C). The second cluster contained variants of Delta_B.1.617.2, Alpha_B.1.1.7, and Beta_B.1.351. When specific BCOV_S1_CTDs’ PDB data were used for structural classification, it showed that the Alpha_B.1.1.7 variant was deviating. The Omicron variants evolved from the Delta_B.1.617.2 and Beta_B.1.351 variants (Figure 2C).

## 3. B-Cell Epitope Prediction in the BCOV_S1_CTD of the S Protein

The S protein functions as a bridge to link to the receptors on the host cell, fusing the viral and host cell membranes and ultimately allowing the virus to enter the host cell [11]. The S protein is an I-type transmembrane glycoprotein that contains the transmembrane domain (TM), ectodomain, and CT domain. The ectodomain comprises two subunits (S1 and S2): the N-terminal domain (NTD) and BCOV_S1_CTD are in the S1 subunit, whereas the fusion peptide (FP) and heptad repeat (HR) domains 1 and 2 are located in the S2 subunit. While the S2 subunit completes the viral fusion and entrance task, BCOV_S1_CTD is in command of attaching to the angiotensin-converting enzyme 2 (ACE2) receptor of host cells [5]. Thus, the S protein is a critical target of SARS-CoV-2 vaccination to prevent COVID-19 and trigger viral growth and transmission. RBD has been identified as a primary target of efficient neutralizing antibodies in previous investigations on SARS-CoV.

The immuno-bioinformatic tools from the IEDB and related resources were used to predict B-cell epitopes in the BCOV_S1_CTD of the S protein from the major variants of SARS-CoV-2 and compare changes in the likely epitope sites from dominant and rare mutations of the S protein. It was observed that different S-protein mutations might have varying effects on those proteins’ putative effective epitopes.

The defense against viral infection heavily depends on humoral immunity. The B-cell epitopes of the S protein, typically present on the viral surface as unprocessed natural antigen molecules, are recognized by the B-cell receptor (BCR) or neutralizing antibodies. We screened the S-protein sequence using the BepiPred-2.0 prediction tool on the IEDB server to identify the probable linear B-cell epitopes. The distribution of B-cell linear epitopes identified in major variants is depicted in Figure 3 [21,22]. Most of the B-cell epitopes were found on RBD domains of the S protein (BCOV_S1_CTD). Subsequently, using the Vaxijen 2.0 tool to analyze antigenicity and the Emini Surface Accessibility Prediction tool to analyze accessibility, the effective epitopes were identified (Figure 3A–D) [21].

The accompanying Table 2 provides the prediction results of the linear B-cell epitope for 10 variants. Additionally, we analyzed epitope alterations by comparing them with the epitopes from the prototype BCOV_S1_CTD (Table 2). Initially, two effective epitopes were identified in the Delta_B.1.617.2 variant out of the five expected epitopes. In contrast, of the five predicted epitopes for the BA.2.13 variant, four were proven to be effective (Table 2). In Table 2, the possible B-cell linear epitopes and their positions, sequences, average scores, and antigenicity are given. The findings of this study revealed that the BA.2.13 variant has four RBD domain epitopes: PFFAFK (40–25), IRGNEVSQIAPGQTGNIADYNYKLPD (69–94), KLDSKVGGNYNYMYRLFRKSNLKPFERD (107–134), and STEIYQAGNKPCNGVAGFNCYFPLRSYGFRPTYGVGHQ (136–173), which have more significant antigenicity and accessibility. Mutations in B-cell epitopes for the neutralizing antibody may result from mutations in the BCOV_S1_CTD of the S protein, affecting its structure and function. Figure 3A,E depict the B-cell linear epitope (Epi_C), which is largely conserved across all Omicron variants. Between all the anticipated B-cell linear epitopes, Epi_A has the highest antigenicity (Figure 3E). Our study also reveals that the Epi_C of Omicron variants has greater antigenicity than that of other variants; this might happen due to some mutations such as D72N, R75S, and K84N in the identified BCOV_S1_CTD.

We compared the predicted epitopes of the major mutants, examined the correlation between epitope changes caused by different mutations, and assessed the impact of mutation on B-cell epitopes to examine the effects of the aforementioned common mutations of BCOV_S1_CTD [23]. The B-cell linear epitopes of the Delta_B.1.617.2, Alpha_B.1.1.7, and Beta_B.1.351 variants were almost identical (Table 2) due to a rare mutation in the BCOV_S1_CTD of its S protein, which was strongly confirmed based on the MSA analysis of these variants (Figure 2). A mutation in the BCOV_S1_CTD of the S protein, however, caused differences in the B-cell linear epitopes in every Omicron variant (B.1.1.529 to BA.4). On the size or position of B-cell epitopes, it was found that, although certain alterations had little or no impact, some modifications had a considerable impact. The most significant finding is that the most common mutation only slightly changed the accessibility and antigenicity of the B-cell epitopes on the S protein.

## 4. Global Vaccine Coverage

As of 21 April 2023, there have been 6.88 million confirmed deaths and 676 million confirmed cases of infection with SARS-CoV-2, which causes COVID-19 [2]. In Asia and the rest of the world, newly confirmed COVID-19 fatalities per million people (7-day rolling average) are increasing again (Figure 4). Since the beginning of the pandemic, virus transmission and mortality have decreased through a variety of strategies, including preventative actions taken by individuals, such as social withdrawal, wearing face masks, prohibiting public gatherings of large numbers of people, and placing travel restrictions on affected areas [24]. Governments are looking to vaccination as a key component of responding to the pandemic following the successful development, assessment, and production of several vaccines.

We need timely cross-national statistics to comprehend the scope and pace of vaccination implementation. A freely available worldwide dataset on given vaccines is available in the World in Data COVID-19 vaccination dataset (https://ourworldindata.org/covid-vaccinations) (accessed on 21 April 2023). It has been frequently updated and covers the whole period beginning on 2 December 2020, when the initial vaccination data were released [25]. The total number of COVID-19 vaccinations administered in each country, broken down into first and second doses, is tracked in this dataset when official statistics are available. Daily vaccination rates and population-adjusted figures are also listed. Users may compare rollout rates between nations, understand the scope and pace of vaccine rollouts relative to population, and assess the priorities for countries with one- and two-dose schedules.

The first reports of COVID-19 vaccines outside of clinical trials were reported on 13 December 2020, in the UK. The global vaccination time series since then are shown in this dataset. Notably, 13.37 billion doses have been administered globally as of 21 April 2023. According to a comparison, the statistics of the world, Asia, and India, the total number of people who completed the COVID-19 vaccination protocol (all doses) is 5 billion worldwide, while in Asia, this number is more than 3 billion (Figure 5A,C). As of 21 April 2023, fewer daily COVID-19 vaccine doses (7-day rolling average; all doses, including boosters, are counted individually) have been administered in India than in the world (Figure 5B). At least one dose of an approved vaccine has been administered to more than 70% of the world’s population. A single day sees the administration of more than 270,000 doses of vaccination [25]. In low-income nations, more than 29% of people have taken at least one dosage. This has drawn attention to important vaccination inequalities throughout the world.

As of 21 April 2023, in India, 155.70 doses have been administered per 100 people, whereas 167.6 doses per 100 people have been administered globally, and 190.7 doses per 100 people have been administered in Asia (Figure 6A). Currently, vaccination coverage in low-income nations has significantly grown. By 21 April 2023, at least one dosage had been administered to more than 80% of Asia (Figure 6B). In Asia and worldwide, the cumulative percentage of people who have completed their initial immunization protocol has increased to 70% and 65%, respectively (Figure 6C). As of 21 April 2023, the cumulative number of booster doses administered per 100 people ranges from 40 per 100 in the case of Asia, 35 per 100 in the world, and 16 per 100 in the case of India (Figure 6D). According to the aforementioned facts, vaccination rates have significantly grown worldwide.

## 5. Novel COVID-19 Therapeutic Strategies

The COVID-19 vaccine was developed after extensive studies to shield individuals from SARS-CoV-2 infection. While 199 vaccines are still in preclinical development phases, at least 183 potential vaccines have started clinical testing [26]. Nucleic acids, protein subunits, virus-like particles, live attenuated and inactivated viruses, and replicating and non-replicating viral vectors have all been used in vaccine development strategies [27]. The vaccines that the World Health Organization (WHO) has approved were created using a range of techniques and have varied levels of efficacy. Given the fact that the S protein of SARS-CoV-2 is essential for receptor binding, the full-length S proteins or its key components, such as its receptor-binding domain (RBD), have been employed as the major target antigen for nucleic acid vaccine candidates [28]. However, newly developing SARS-CoV-2 strains have modified these antigens. Twelve vaccine candidates are undergoing clinical testing and have been given the go-ahead by several national regulatory organizations. The bulk (32%) of the recommended vaccine types are protein-based vaccines, with 21 potential vaccines now in Phase III and one in Phase IV. The European Medicines Agency (EMA) approved the first protein-based vaccine NVX-CoV2373 from Novavax (Gaithersburg, MD, USA), with an effectiveness of 89.7%. In order to prevent SARS-CoV-2, this drug was approved in December 2021 [29]. RNA-based vaccines are the second-largest type of vaccine in development, which accounts for around 24% [26]. The first of these vaccines to be approved by the WHO for use in an emergency to control COVID-19 was Pfizer-BioNTech’s BNT162b2 (Manhattan, NY, USA) [30]. Surprisingly, 66 days after the SARS-CoV-2 sequence was made public, Moderna’s mRNA-1273 commenced its first US clinical trial (Table 3). These two products—the first RNA vaccines approved for clinical use—have sufficiently demonstrated the advantages and potential applications of RNA-based immunizations [31]. For the COVID-19 vaccine competition, 222 vaccine candidates were developed by teams from 79 countries, and more than 67% of them have begun Phase II trials. After the protein subunit (32%) and RNA (24%), inactivated viruses (13%), non-replicating viral vectors (13%), and DNA (9%) were found as the most effective vaccine candidates studied in clinical studies [26]. Since the pandemic’s start, a lot has been learned about the various vaccine kinds, their efficacy, and their safety. Today, a major issue with vaccine use is equitable access to effective immunization. In the following sections, we review the theory and design of the three main vaccine types (protein subunit, adenoviral vector, and mRNA) and the efficacy of these vaccines against the various SARS-CoV-2 variants (Table 3).

### 5.1. Comparative Effectiveness of Vaccines against COVID-19

#### 5.1.1. Vaccines with the Protein Subunit

Protein subunit vaccines generate immune responses to one or more isolated viral proteins rather than the whole viruses. Since these vaccines contain no live organisms, the risk of pathogenicity is completely reduced, and vaccination can even be administered to patients with impaired immune systems [32]. Protein subunit vaccines are also a proven technique with a long application history, and the products are comparatively stable during storage and transit. However, protein subunit vaccinations often have a limited capacity to induce an immune response and may need adjuvants and numerous doses to achieve protective immune responses [33]. Recombinant protein creation and production need a lengthy and intricate procedure. Some of the protein subunit vaccines authorized for clinical use are the human papillomavirus, hepatitis B, and influenza vaccines [34,35,36]. SARS-CoV-2 protein subunit vaccines frequently target full-length S proteins or their antigenic components, such as the S1 subunit and RBD [37]. While 17 protein subunit vaccines against SARS-CoV-2 have been granted emergency use permits, 55 candidates are presently undergoing clinical testing. One of these is NVX-CoV2373, which has been approved in 37 nations and is regarded as one of the finest protein subunit vaccines for SARS-CoV-2. A two-dose course of the NVX-CoV2373 vaccine enabled good protection against the B.1.1.7 variant and offered 89.7% protection against SARS-CoV-2 infection, according to Phase III clinical studies [38].

#### 5.1.2. Vector-Based Adenovirus COVID-19 Vaccines

The majority of the viral vectors utilized for SARS-CoV-2 vaccinations are, by far, adenoviruses. These non-enveloped icosahedral viruses containing DNA were first discovered in the 1950s [39] and feature a capsid with a diameter of around 90 nm. Adenoviral vectors effectively transfer the desired genes into the host cells, which is accompanied by host immunological responses that prevent vector transduction and transgenic expression. Adenoviral vectors constitute an excellent vaccine platform because of their high immunogenicity and transitory gene expression, which eliminates the need for additional adjuvants. Since adenoviral vector-based vaccines are easier to manufacture and develop more quickly than protein or subunit vaccines, they were identified as options for vaccine platforms after the SARS-CoV-2 genome sequence was determined in January 2020. So far, four adenoviral vector-based vaccines have been given the go-ahead by various regional organizations. Human adenovirus 5 (Ad5) was once the most common viral vector utilized to create vaccines. Ad5-nCoV, the CanSino Ad5 vector-based COVID-19 vaccine, was created and approved in China [40]. Ad5-nCoV is 57.5% successful at treating symptomatic COVID-19 infection with a single dose [41].

The AstraZeneca ChAdOx1 nCoV-19 vaccine (AZD1222; brand name, Vaxzevria) (Sweden) was first given conditional authorization by the European Medicines Agency (EMA). In non-human primates exposed to SARS-CoV-2, the AZD1222 vaccine efficiently decreased lung damage after a single dose (Table 3) [42]. According to two Phase III trials [43,44], the overall immunization efficacy in those who received two standard doses was estimated to be about 70%. It is significant to note that AZD1222-induced antibodies might encourage complement deposition, antibody-dependent NK cell activation, and neutrophil/monocyte phagocytosis [45], all of which may help contain SARSCoV-2 infection. Similarly, the Janssen COVID-19 vaccine (Ad26.COV2.S) was initially authorized by the US Food and Drug Administration (FDA). Beginning in July 2020, Phase I/II trials have demonstrated good immunogenicity and tolerance [46]. According to Phase III research, the Janssen COVID-19 vaccine is 66.9% effective for COVID-19 and provides higher protection (76.7%) over severe-to-critical symptoms 14 days following vaccination [47]. In a second instance, the Gamaleya Research Institute used the heterologous prime–boost strategy of Ad26 and Ad5 (each encoding the full-length S protein) to create the Russian Sputnik V vaccine. An interim analysis of Phase III clinical research in Russia demonstrated 91.6% efficacy against COVID-19 [48].

#### 5.1.3. Vaccines Based on mRNA

The scientific community was forced to develop vaccines swiftly while maintaining their safety and efficacy due to the development of the COVID-19 pandemic. Due to the straightforward, trustworthy, and adaptable technological method utilized to create new candidate vaccines, mRNA-based vaccines are undoubtedly the greatest option for fast development. COVID-19 vaccines were developed and tested using this technology, which has so far outperformed the more time-consuming conventional methods of vaccine creation. Additionally, mRNA vaccines have a high level of efficacy due to their quick uptake and expression, which is supported by their formulation’s non-infectious and non-integrating characteristics. The ability to produce mRNA vaccines at a low cost may be their most significant benefit [49].

### 5.2. Therapeutic Antibodies with High Efficacy

Finding therapeutic antibodies for the Omicron variation is a significant difficulty for researchers since these antibodies’ efficacy in neutralizing the Omicron variant is compromised [50,51]. As a result, various researchers have attempted to examine the efficacy of therapeutic antibodies against the Omicron form over time. Omicron antibodies such as bamlanivimab (LY-CoV555), tixagevimab (COV2-2196), imdevimab (REGN10987), casirivimab (REGN10933), and sotrovimab precursors (S309) were evaluated by Takashita et al. They also tested a wide range of monoclonal antibody combinations such as tixagevimab and cilgavimab, imdevimab, casirivimab, etesevimab, and bamlanivimab. Additionally, it was found that these monoclonal antibody mixtures could neutralize both the wild strain and the Delta and Alpha strains. Combining bamlanivimab and etesevimab indicated a concurrently diminished neutralizing activity against the Gamma variant. These combinations can no longer neutralize the Beta and Omicron versions [52]. The combination of casirivimab and imdevimab also demonstrated activity against the Gamma and Beta strains. The Omicron variant was still unaffected by this combo. On the other hand, the cilgavimab–tixagevimab combination was shown to have significant neutralization power against the Beta, Gamma, and Omicron variants [52]. However, it was established that the Omicron pseudotype was unresponsive to many monoclonal antibodies [53]. The therapeutic antibodies against Omicron have been identified with the aid of several in silico investigations. Shah and Woo proposed that sotrovimab (GSK, S203 mAb) and Evusheld (AstraZeneca mAbs) may be used in conjunction to successfully suppress the Omicron variant in this region [54].

Additionally, researchers have tried to understand the relationship between the Omicron spike protein and the neutralizing antibodies (nABs). This could help us understand the special interaction mechanisms that these antibodies have. Recent investigations have revealed that ZCB11 may be a possible antibody for an Omicron. Zhou et al. described the connection between ZCB11 and the spike protein of the Omicron version (PDB id: 7XH8). In their study, ZCB11 was shown to target the viral RBD domain and the spike protein of the Omicron SARS-CoV-2 variants [55].

### 5.3. Novel Antiviral Drugs against SARS-CoV-2

Natural products are essential to patient care because of their distinctive structural, chemical, and biological variety. They are a traditional source for contemporary pharmaceutical discovery and possible therapeutic leads [56,57]. Remdesivir, an antiviral medicine, has received significant recognition for its capacity to regulate viral infection and was approved by the FDA for treating COVID-19 patients suffering from pneumonia at the same time as an oxygen supply crisis [58]. It serves as a broad-spectrum phosphoramidite prodrug and adenosine nucleotide analog that may target a variety of viruses, including coronaviruses. A Janus kinase–STAT signaling inhibitor (JAK-STAT), baricitinib prevents the production of cytokines from rising and has antiviral and anti-inflammatory effects by blocking clathrin-mediated endocytosis [59]. Similar to this, molnupiravir, a recently FDA-approved antiviral medication against SARS-CoV-2 infection, is known to work by focusing on the viral polymerase and tricking it into incorporating adenosine or guanosine during viral replication, ultimately causing an accumulation of harmful errors that renders the virus non-infectious [60,61]. Although investigations have shown that the sequences important for viral RdRp activity are still conserved in the earliest and novel SARS-CoV-2 variants, it is improbable that novel VOCs would interfere with the effectiveness of such antiviral medications. Additionally, other findings support the unrestricted use of the latest FDA-approved Paxlovid antiviral medication against the current VOCs, particularly the most recent omicron variants [62]. Therefore, given the information above, it may be logical to say that the action of such antivirals against developing VOCs may not be hampered [63].

Furthermore, there has been considerable advancement in our understanding of natural compounds that are effective against COVID-19. A crucial component of traditional herbal therapy of *Ginkgo biloba* (EGb) is the phenolic compound ginkgolic acid [64,65]. According to an investigation, “the half-maximal inhibitory concentrations (IC_50_) of ginkgolic acid against SARS-CoV-2 Mpro and SARS-CoV-2 PLpro are 1.79 µM and 16.3 µM, respectively” [66]. Another intriguing target that controls the viral genome replication is SARS-CoV-2’s RNA-dependent RNA polymerase (RdRp) [67]. A gallotannin called corilagin, which is a non-nucleoside inhibitor, has been identified in the medicinal herb *Phmllanthi fructus* [68]. Corilagin has been shown to block the conformational shift of RdRp and limit SARS-CoV-2 replication, with an EC_50_ value for SARS-CoV-2 infection of 0.13 µM in a concentration-dependent manner [69]. In the conventional herbal remedy *Stephania cephalantha Hayata*, the bisbenzylisoquinoline alkaloid cepharanthine may be extracted [70]. According to Ohashi et al. (2021) [71], this alkaloid inhibits SARS-CoV-2 entrance in vitro at an IC_50_ of 0.35 µM without showing any signs of toxicity (selectivity index, [SI] > 70), thus highlighting the possibility of cepharanthine as a therapy option for SARS-CoV-2 infection. In a recent study, “nelfinavir was shown to have minimal toxicity (SI = 3.7) and effective at preventing SARS-CoV-2 Mpro infection (IC50 = 3.3 µM). The SARS-CoV-2 S protein was also completely inhibited by nelfinavir at a concentration of 10 M, with no signs of harm to cells” [72]. Several studies have shown that nelfinavir and cepharanthine have anti-SARS-CoV-2 efficacy in vitro by inhibiting SARS-CoV-2 Mpro and partially S protein, respectively [71]. Further study is required to determine whether cepharanthine and nelfinavir have synergistically enhanced activity for treating SARS-CoV-2 infection in patients, taking into account all of the above factors, including their critical importance in both in vitro and animal models of anti-SARS-CoV-2 infection as well as mathematical prediction modeling. Following testing with SARS-CoV-2 and animal models, these inhibitors may be developed into therapeutic candidates [73]. More study is required to determine whether they have anti-SARS-CoV-2 efficacy in vivo.

## 6. Emerging SARS-CoV-2 Variants

SARS-CoV-2 continues to evolve, posing a significant risk to world health due to its increased infectivity and rapid transmission. The WHO categorizes them into VOIs and VUMs in order to more accurately analyze the effects of various variants and promote preventative or medicinal countermeasures (https://www.who.int/activities/tracking-SARS-CoV-2-variants) (accessed on 30 March 2023) [74]. There are currently one VOI (XBB.1.5; Table 4A) and seven VUMs (BA.2.75, CH.1.1, BQ.1, XBB, XBB.1.16, XBB.1.9.1, and XBF; Table 4B). The appearance of Omicron has now prompted more focus and caution. The S protein is the region where most of the mutations in Omicron are located [10], and there appears to be a propensity to accumulate mutations that facilitate immunological escape [75,76,77]. According to a model, Omicron is around ten times as infectious as the original virus or twice as infectious as the Delta version. Based on Figure 2, the crucial mutation sites in the SARS-CoV-2 genome that govern its virulence and ability to spread are listed below, which presents new opportunities for creating medications to treat the major emerging variants.

## 7. Conclusions and Future Perspectives

The binding of the SARS-CoV-2 S protein to the ACE2 receptor is an essential step for the virus to invade the human body. The BCOV_S1_CTD of the S protein is the region in which most of the mutation sites are currently being investigated [16,78]. The present SARS-CoV-2 genome site mutations are all an outcome of drug screening and natural selection, which shows how the virus has adapted to different therapies. However, earlier drugs continue to be therapeutically effective against SARS-CoV-2 variants [1] despite the several changes that made the virus more contagious [1]. Currently, the majority of COVID-19 vaccines were developed to target S proteins to stimulate the production of neutralizing antibodies [79], and the majority of vaccines undergoing Phase III clinical trials are based on the early S protein [80]. However, due to its fast and extensive worldwide distribution, SARS-CoV-2 can change and develop. Our analysis revealed that almost 10.88% of sites were crucial in amino acid replacement. The S protein’s mutation in BCOV_S1_CTD may impact B-cell epitopes and result in vaccination failure. Therefore, in this investigation, we used immuno-informatic methods to identify probable B-cell epitopes in the BCOV_S1_CTD of the main variations to investigate the effect of mutations on the antigenicity of BCOV_S1_CTD. Most of the B-cell epitopes were found on the S protein’s BCOV_S1_CTD. This study also revealed that Epi_C is the most potent and conserved epitope among the epitopes of the Omicron variants. This would facilitate the wide use of SARS-CoV-2 prototype vaccines, even in locations with a high prevalence of the virus and an abundance of mutant strains [81]. Within a year, scientists developed several COVID-19 vaccines that are incredibly effective. The challenge now is whether vaccines can be distributed rapidly and equally worldwide to match the speed at which they were produced. One of the best protein subunit vaccines for SARS-CoV-2, NVX-CoV2373, has received approval from 37 countries. Phase III clinical studies showed that an effective two-dose course of the NVX-CoV2373 vaccine provided 89.7% protection against SARS-CoV-2 infection and afforded significant protection against the B.1.1.7 variant. Due to their rapid absorption and expression, mRNA vaccines have a high level of effectiveness, which is reinforced by their formulation’s non-infectious and non-integrating properties.

On the other hand, it has been demonstrated that the Omicron antibodies cilgavimab and tixagevimab exhibit potent neutralization activity against the Beta, Gamma, and Omicron variants. Therefore, it is probably appropriate to say that the action of these antivirals may be unaffected by the newly developing VOCs, which would be desirable for developing the next generation of antiviral therapies. The above data show that vaccine coverage has enormously increased throughout the world.

Over time, efforts have been made to develop next-generation and mutation-proof vaccines [51,82]. The Omicron variant and its subvariant have been shown to have many mutations. Hybrid immunity has recently been found to increase immune defense against SARS-CoV-2 and other VOCs [51]. Therefore, we must explore the possibility of hybrid immunity for protection against Omicron. Overall, the successful control of the current pandemic might rely on the continuous devotion of researchers, vaccine manufacturers, national regulators, and the whole public health system.

## Figures and Tables

**Figure 1 viruses-15-01234-f001:**
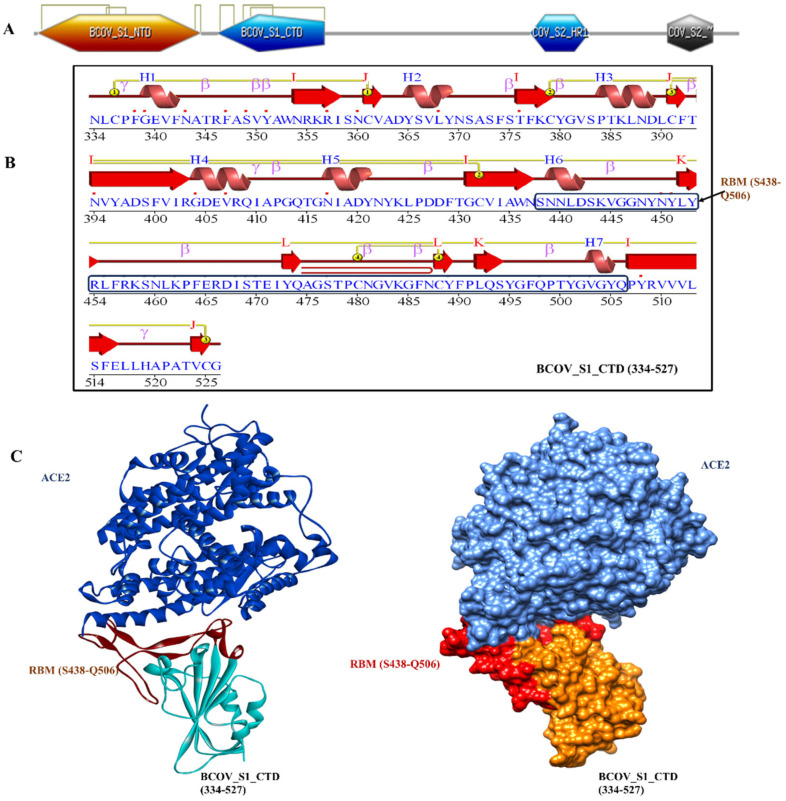
The structure of BCOV_S1_CTD bound to ACE2: (**A**) overall domain profile of the SARS-CoV-2 spike monomer. FP, fusion peptide; HR1, heptad repeat 1; HR2, heptad repeat, NTD: N-terminal domain and BCOV_S1_CTD; (**B**) sequence and secondary structures of BCOV_S1_CTD and RBM sequence bound to ACE2 are shown in red; (**C**) the BCOV_S1_CTD core is shown in cyan and RBM in red.

**Figure 2 viruses-15-01234-f002:**
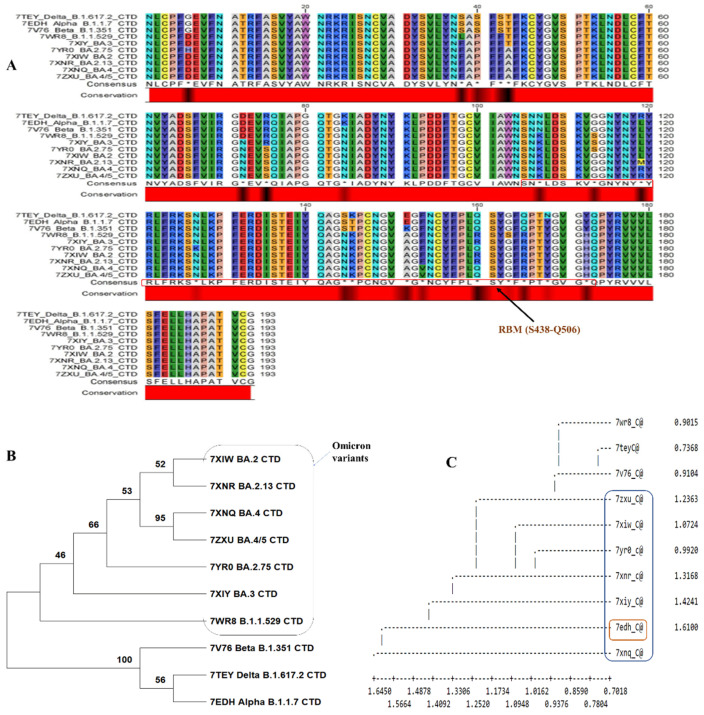
Analysis of the BCOV_S1_CTD sequence: (**A**) the MSA of BCOV_S1_CTD was performed using Crustal Omega, with sequence alignment showing BCOV_S1_CTD in major variants of COVID-19. The critical residues for binding between SARS-CoV RBM and human ACE2 protein are indicated in red boxes; (**B**) phylogenetic trees of the SARS-CoV-2-related lineage estimated from the entire BCOV_S1_CTD region. The results of 1000 bootstrap replicates’ worth of branch supports are displayed; (**C**) dendrogram showing the structural alignment of the BCOV_S1_CTD in major variants.

**Figure 3 viruses-15-01234-f003:**
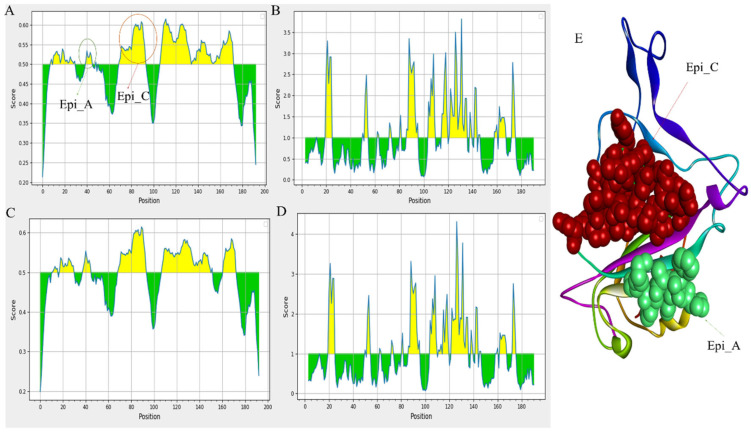
Prediction of B-cell linear epitopes and accessibility analysis of BCOV_S1_CTD of S protein: (**A**) the distribution of all the predicted B-cell linear epitopes using BepiPred-2.0. The displayed possible epitope residues are those with scores above the cutoff (value is adjusted at 0.50) and are highlighted in yellow. Y-axes indicate residue scores, and X-axes exhibit residue positions of the BCOV_S1_CTD of S protein (BA.2.13 variant); (**B**) the surface accessibility analyses using the Emini surface accessibility scale. The residues with scores above the threshold (the default value is 1.00) are predicted to have good accessibility (BA.2.13 variant). The same holds for the (**C**,**D**) of the BCOV_S1_CTD of S protein (BA.2.75 variant); (**E**) the B-cell linear epitope (Epi_C), which is mostly conserved across all the Omicron variants, and the B-cell linear epitope (Epi_A), which has the maximum antigenicity among all the predicted B-cell linear epitopes.

**Figure 4 viruses-15-01234-f004:**
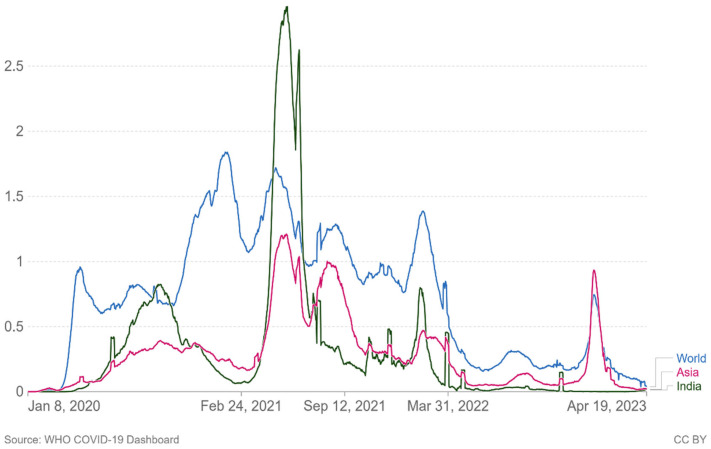
Cumulative number of newly confirmed COVID-19 deaths per million people (7-day rolling average). Due to varying protocols and challenges in the attribution of the cause of death, the number of confirmed deaths may not accurately represent the true number of deaths due to COVID-19.

**Figure 5 viruses-15-01234-f005:**
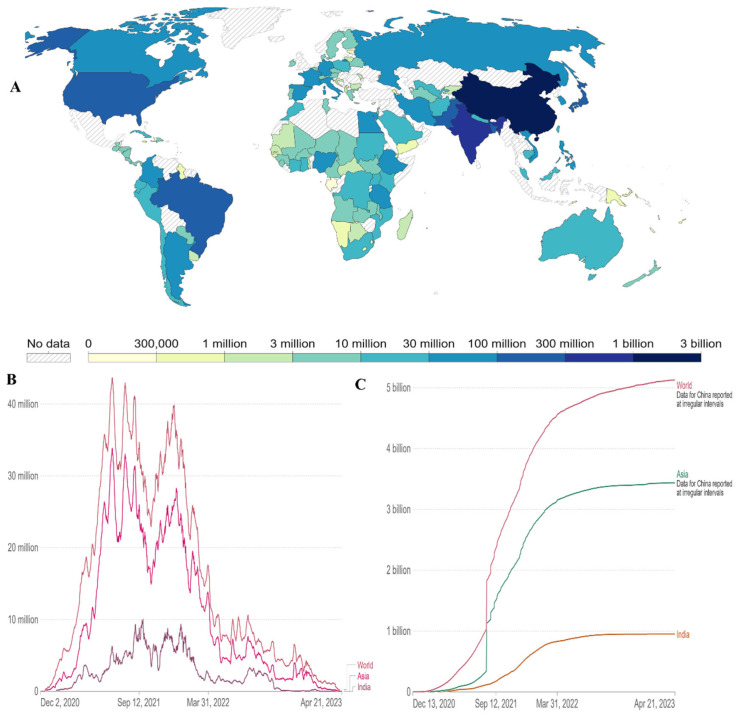
(**A**) The cumulative number of people who completed the COVID-19 initial vaccination protocol administered by country until 21 April 2023; (**B**) daily COVID-19 vaccine doses administered (7-day rolling average, all doses, including boosters, are counted individually); (**C**) comparative analysis of the number of people (world, Asia, and India) for whom the COVID-19 vaccination protocol (all doses) was administered by country till 21 April 2023. Data are available online: https://ourworldindata.org/coronavirus (accessed on 21 April 2023).

**Figure 6 viruses-15-01234-f006:**
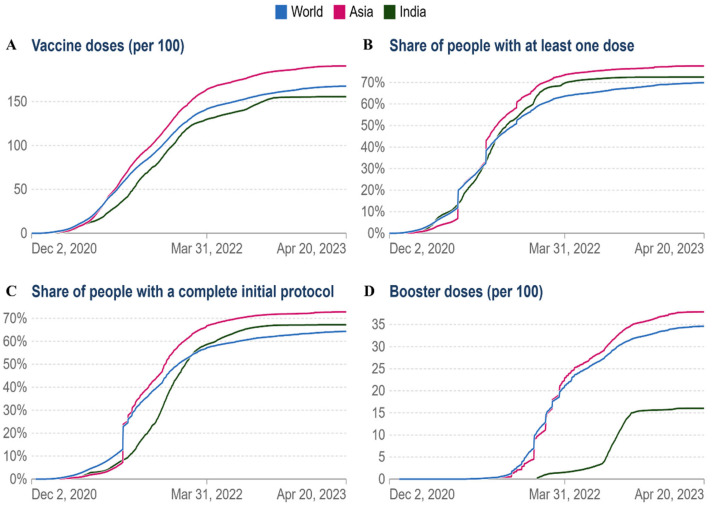
Comparative analysis of vaccine doses administered per 100 people in the world, Asia, and India: (**A**) the cumulative total vaccine doses administered per 100 people over time; (**B**) share of people with at least one dose; (**C**) people with a full initial vaccination protocol; (**D**) booster doses administered per 100 people.

**Table 1 viruses-15-01234-t001:** The coronavirus spike glycoprotein domain profile from SPIKE_SARS2 (UniProt ID: P0DTC2).

S.No.	ScanProsite ID	Name	Start Position	End Position	Details
**1.**	PS51922	BCOV_S1_NTD	9	303	Betacoronavirus spike (S) glycoprotein S1 subunit N-terminal (NTD) domain
**2.**	PS51921	BCOV_S1_CTD	334	527	Betacoronavirus spike (S) glycoprotein S1 subunit C-terminal (CTD) domain
**3.**	PS51923	COV_S2_HR1	896	1001	Coronavirus spike (S) glycoprotein S2 subunit heptad repeat 1 (HR1) region
**4.**	PS51924	COV_S2_HR2	1143	1225	Coronavirus spike (S) glycoprotein S2 subunit heptad repeat 2 (HR2) region

**Table 2 viruses-15-01234-t002:** BipiPred linear epitopes 2.0 prediction program on the IEDB server was used to predict the B-cell epitope in the BCOV_S1_CTD of S protein, along with their start and end position, average score, and antigenicity score with Vaxijen 2.0 tool.

Proteins/CTD-Domain	Average Score	Position	Sequences	Antigenicity
**Delta_B.1.617.2**	0.503	11–30	ATRFASVYAWNRKRISNCVA	0.2689 (NA)
		39–45	ASFSTFK	0.0865 (NA)
		69–94	IRGDEVRQIAPGQTGKIADYNYKLPD	0.9322 (A)
		107–152	NLDSKVGGNYNYRYRLFRKSNLKPFER DISTEIYQAGSKPCNGVEG	0.3435 (NA)
		161–173	SYGFQPTNGVGYQ	0.7632 (A)
**Alpha_B.1.1.7**	0.503	11–30	ATRFASVYAWNRKRISNCVA	0.2689 (NA)
		39–45	ASFSTFK	0.0865 (NA)
		69–94	IRGDEVRQIAPGQTGKIADYNYKLPD	0.9322 (A)
		107–152	NLDSKVGGNYNYRYRLFRKSNLKPFER DISTEIYQAGSKPCNGVEG	0.3435 (NA)
		161–173	SYGFQPTNGVGYQ	0.7632 (A)
**Beta_B.1.351**	0.501	11–30	ATRFASVYAWNRKRISNCVA	0.2689 (NA)
		39–45	ASFSTFK	0.0865 (NA)
		69–94	IRGDEVRQIAPGQTGKIADYNYKLPD	0.9322 (A)
		107–152	NLDSKVGGNYNYRYRLFRKSNLKPFER DISTEIYQAGSKPCNGVEG	0.3435 (NA)
		161–173	SYGFQPTNGVGYQ	0.7632 (A)
**B.1.1.529**	0.501	8–30	VFNATRFASVYAWNRKRISNCVA	0.2656 (NA)
		38–45	LAPFFTFK	1.0698 (A)
		69–94	IRGDEVRQIAPGQTGNIADYNYKLPD	0.9322 (A)
		108–134	LDSKVSGNYNYLYRLFRKSNLKPFERD	0.3225 (NA)
		137–173	TEIYQAGNKPCNGVAGFNCYFPLRSYSFRPTYGVGHQ	0.5562 (A)
**BA.2**	0.502	8–30	VFNATRFASVYAWNRKRISNCVA	0.2656 (NA)
		39–46	APFFAFKC	1.2004 (A)
		69–94	IRGNEVSQIAPGQTGNIADYNYKLPD	1.0563 (A)
		108–152	LDSKVGGNYNYLYRLFRKSNLKPFER DISTEIYQAGNKPCNGVAG	0.2073 (NA)
		156–173	YFPLRSYGFRPTYGVGHQ	0.4765 (A)
**BA.2.13**	0.499	11–30	ATRFASVYAWNRKRISNCVA	0.2689 (NA)
		40–45	PFFAFK	1.9601 (A)
		69–94	IRGNEVSQIAPGQTGNIADYNYKLPD	1.0563 (A)
		107–134	KLDSKVGGNYNYMYRLFRKSNLKPFERD	0.4904 (A)
		136–173	STEIYQAGNKPCNGVAGFNCYFPLRSYGFRPTYGVGHQ	0.4726 (A)
**BA.2.75**	0.494	12–16	TRFAS	0 (NA)
		18–30	YAWNRKRISNCVA	0.3936 (NA)
		38–45	FAPFFAFK	1.1148 (A)
		69–94	IRGNEVSQIAPGQTGNIADYNYKLPD	1.0563 (A)
		108–150	LDSKVSGNYNYLYRLFRKSKLKPFER DISTEIYQAGNKPCNGV	0.0655 (NA)
		162–173	YGFRPTYGVGHQ	0.7884 (A)
**BA.3**	0.497	11–16	ATRFAS	−0.151 (NA)
		18–30	YAWNRKRISNCVA	0.3936 (NA)
		38–45	FAPFFAFK	1.1148 (A)
		69–94	IRGNEVSQIAPGQTGNIADYNYKLPD	1.0563 (A)
		108–150	LDSKVSGNYNYLYRLFRKSKLKPFER DISTEIYQAGNKPCNGV	0.0655 (NA)
		162–173	YGFRPTYGVGHQ	0.7884 (A)
**BA.4**	0.498	11–16	ATRFAS	−0.151 (NA)
		18–30	YAWNRKRISNCVA	0.3936 (NA)
		39–45	APFFAFK	1.2513 (A)
		69–94	IRGNEVSQIAPGQTGNIADYNYKLPD	0.9322 (A)
		108–150	LDSKVSGNYNYLYRLFRKSKLKPFER DISTEIYQAGNKPCNGV	0.0655 (NA)
		162–173	YGFRPTYGVGHQ	0.7884 (A)
**BA.4/5**	0.498	11–16	ATRFAS	−0.151 (NA)
		18–30	YAWNRKRISNCVA	0.3936 (NA)
		39–45	APFFAFK	1.2513 (A)
		69–94	IRGNEVSQIAPGQTGNIADYNYKLPD	0.9322 (A)
		108–150	LDSKVSGNYNYLYRLFRKSKLKPFER DISTEIYQAGNKPCNGV	0.0655 (NA)
		162–173	YGFRPTYGVGHQ	0.7884 (A)

**Table 3 viruses-15-01234-t003:** Different types of vaccine in clinical trials against major variants of COVID-19. Available online: https://clinicaltrials.gov (accessed on 20 April 2023).

SL NO.	Vaccine	Company Name	Variant	Trial Number	Phase	Recruitment Status	Number of Participants	Study Completion Date	Reference Clinical Trial Link
**1.**	BNT162b2	Pfizer-BioNTech	Alpha	NCT04368728	III	Operative	47,079	10 February 2023	https://clinicaltrials.gov/ct2/show/NCT04368728
**2.**	mRNA-1273	ModernaTX, Inc.	Beta	NCT04470427	III	Operative	30,000	29 December 2022	https://clinicaltrials.gov/ct2/show/NCT04470427
**3.**	ChAdOx1 (AZD1222)	AstraZeneca	Gamma	NCT04516746	III	Operative	32,459	10 February 2023	https://clinicaltrials.gov/ct2/show/NCT04516746
**4.**	Ad26.COV2. S	Johnson & Johnson	Delta	NCT04505722	III	Operative	44,325	31 March 2023	https://clinicaltrials.gov/ct2/show/NCT04505722
**5.**	CoronaVac	SinoVac Biotech	Omicron	NCT04456595	III	Operative	12,688	February 2022	https://clinicaltrials.gov/ct2/show/NCT04456595
**6.**	mRNA-127 3.214	Sheba Medical Center	Omicron	NCT05383560	II	Not recruited yet	150	July 2023	https://clinicaltrials.gov/ct2/show/NCT05383560
**7.**	COVID-19 bivalent vaccine	Pfizer-Bio NTech	Omicron	NCT04977479	II	Operative	17	22 February 2023	https://clinicaltrials.gov/ct2/show/NCT04977479
**8.**	Bivalent mRNA COVID-19 vaccine	NIAID	Omicron	NCT05077254	II	Recruiting	400	June 2024	https://clinicaltrials.gov/ct2/show/NCT05077254
**9.**	Bivalent booster of mRNA-based COVID-19 vaccine	NIAID	Omicron	NCT05518487	II	Not recruited yet	80	15 July 2024	https://clinicaltrials.gov/ct2/show/NCT05518487
**10.**	SCTV01E	Sinocelltech Ltd.	Omicron	NCT05308576	III	Not recruited yet	10,000	October 2024	https://clinicaltrials.gov/ct2/show/NCT05308576

**Table 4 viruses-15-01234-t004:** (**A**) List of currently circulating VOIs (as of 30 March 2023); (**B**) list of currently circulating VUMs. https://www.who.int/activities/tracking-SARS-CoV-2-variants. (accessed on 30 March 2023).

(A) Pango Lineage	Next Strain Clade	Genetic Features	Earliest Documented Samples	Date of Designation and Risk Assessments
**XBB.1.5**	23A	Recombinant of BA.2.10.1 and BA.2.75 sublineages, i.e., BJ1 and BM.1.1.1, with a breakpoint in S1. XBB.1 + S: F486P (similar spike genetic profile as XBB.1.9.1)	05-01-2022	11 January 2023 XBB.1.5 Rapid Risk Assessment, 11 January 2023 XBB.1.5 Updated Risk Assessment, 24 February 2023
(B) Pango Lineage	Next Strain Clade	Genetic Features	Earliest Documented Samples	Date of Designation and Risk Assessments
**BA.2.75**	22D	BA.2 + S: K147E, S: W152R, S: F157L, S: I210V, S:G257S, S:D339H, S:G446S, S:N460K, S:Q493R reversion	31-12-2021	06-Jul-2022
**CH.1.1**	22D	BA.2.75 + S: L452R, S: F486S	27-07-2022	08-Feb-2023
**BQ.1**	22E	BA.5 + S: R346T, S:K444T, S:N460K	07-02-2022	21-Sep-2022
**X.B.B.**	22F	BA.2 + S:V83A, S:Y144-, S:H146Q, S:Q183E, S:V213E, S:G252V, S:G339H, S:R346T, S:L368I, S:V445P, S:G446S, S:N460K, S:F486S, S:F490S	13-08-2022	12-Oct-2022
**XBB.1.16**	Not assigned	Recombinant of BA.2.10.1 and BA.2.75 sublineages, i.e., BJ1 and BM.1.1.1 XBB.1 + S: E180V, S: K478R and S: F486P	23-01-2023	22-03-2023
**XBB.1.9.1**	Not assigned	Recombinant of BA.2.10.1 and BA.2.75 sublineages, i.e., BJ1 and BM.1.1.1 XBB.1 + S:F486P (similar spike genetic profile as XBB.1.5)	05-12-2022	30-03-2022
**X.B.F.**	Not assigned	Recombinant of BA.5.2.3 and CJ.1 (BA.2.75.3 sublineage) BA.5 + S:K147E, S:W152R, S:F157L, S:I210V, S:G257S, S:G339H, S:R346T, S:G446S, S:N460K, S:F486P, S:F490S	27-07-2022	08-Feb-2023

## Data Availability

Not applicable.

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
