# Peer review of "Recent Developments and Future Perspectives of Vaccines and Therapeutic Agents against SARS-CoV2 Using the BCOV_S1_CTD of the S Protein"

_viruses, 2023, doi:10.3390/v15061234_

Round 1

Reviewer 1 Report

Comments and Suggestions for Authors

“Novel antiviral therapeutic against SARS-CoV-2 and its Variants: Recent developments and future perspectives” is informative manuscript. The manuscript needs revision before publication. I have the following suggestions for authors to address.

1.     Check the abbreviations throughout the manuscript and introduce the abbreviation when the full word appears the first time in the abstract and the remaining for the text and then use only the abbreviation (For example, VOI, VUMs). Make a word abbreviated in the article that is repeated at least two times in the text, not all words to be abbreviated (For example, FDA).

2.     There is a lack of recent literature citations. For example, in lines 77-78 “The membrane protein (M), nucleocapsid protein (N), spike protein (S) and envelop protein (E) are the four structural proteins. (DOI: 10.1016/j.csbj.2021.08.029)”.

3.     “SARS-CoV-2 virus”---“SARS-CoV-2” (in lines 31, 74, 455); In addition, “Covid-19”---“COVID-19” (in lines 367, 378, 395, 404)

4.     The text should be consistent with the title “Novel antiviral therapeutic against SARS-CoV-2 and its Variants……”. The research and development of drugs to effectively prevent and treat COVID 19 has become a major strategic and social demand. In the manuscript, it is better to include a paragraph describing the “conventional antiviral therapeutic against SARS-CoV-2”. For example, “In the face of this epidemic, small molecule agents (Viruses. 2023, 15, 580), natural products (Biomedicines. 2021, 9, 689; Front. Pharmacol. 2022, 13, 926507), and traditional medicine (Cell & Bioscience. 2021, 11, 100) have also played significant role in treating COVID-19.”

5.     References “52-61” can be removed.

6.     In “Conclusion and future perspectives” section, the conclusion seems very general, lacking the future aspects and major findings. The quality of the conclusion must be improved. In addition, it is better to include a paragraph describing the perspective of what are the desired properties for the next-generation of antiviral therapeutic (with new variants continue to evolve)?

Comments on the Quality of English Language

Minor editing of English language required

Reviewer 2 Report

Comments and Suggestions for Authors

Globally, the article is written in an interesting way and contains a lot of detailed data, but it is framed chaotically.

The title of the article is unfortunate and does not reflect its content. It is recommended to change to something like " Recent developments and future perspectives of vaccines and therapeutic agents against SARS-CoV2 using BCOV_S1_CTD of S protein”

It is necessary to add to the article information about the essential anti SARS-CoV2 drugs currently being developed.

Section numbering needs to be revised. In section 2, the text comes first, then subsection 2.1 appears, there is no expected subsection 2.2, then immediately comes section 3, which is thematically closely related to the content of section 2. It is quite possible to combine them.

Section 6 would be more logical before section 5, as it contains general information on vaccination coverage. Section 7 can be moved to the introduction.

Round 2

Reviewer 2 Report

Comments and Suggestions for Authors

The article is reviced and can be accepted at present form.